# Collaboration-Aware Hybrid Learning for Knowledge Development Prediction

## ABSTRACT

In recent years, the rise of online knowledge management platforms has significantly improved work efficiency in enterprises. Knowledge development prediction, as a critical application within these platforms, enables organizations to proactively address knowledge gaps and align their learning initiatives with evolving job requirements. However, it still confronts challenges in exploring collaborative networks and adapting to ecological situations in working environment. To this end, in this paper, we propose a Collaboration-Aware Hybrid Learning approach (CAHL) for predicting the future knowledge acquisition of employees and quantifying the impact of various knowledge learning patterns. Specifically, to fully harness the inherent rules of knowledge development, we first learn the knowledge co-occurrence and prerequisite relationships with an association prompt attention mechanism to generate effective knowledge representations through a specially-designed *Job Knowledge Embedding* module. Then, we aggregate the features of mastering knowledge and work collaborators for employee representations in another *Employee Embedding* module. Moreover, we propose to model the process of employee knowledge development via a *Hybrid Learning Simulation* module that integrates both collaborative learning and self learning to predict future-acquired job knowledge of employees. Finally, extensive experiments conducted on a real-world dataset clearly validate the effectiveness of CAHL [1].

## CCS CONCEPTS

• **Information systems** → **Data mining**.

## KEYWORDS

Knowledge development, knowledge management system, web mining, content analysis

## 1 INTRODUCTION

In the fast-evolving knowledge economy era, efficient knowledge learning has become a crucial success factor and driving force to achieve sustainable competitive advantage for talent development [20, 42]. Recently, the emergence of online Knowledge Management Systems (KMSs) such as Viva [4] and Slack [30] has assisted employees with knowledge learning and project collaboration. As

---

[1]The code and data are available at https://anonymous.4open.science/r/CAHL.

*WWW '24, MAY 13–17, 2024, Singapore*
© 2024 Association for Computing Machinery.
ACM ISBN 978-1-4503-XXXX-X/18/06...$15.00
https://doi.org/XXXXXXX.XXXXXXX

| Employee | Mastering knowledge |
|----------|---------------------|
| A | Python, PyTorch, Recommendation, … |
| B | Python, API design, Backend development, … |
| C | Image caption generation, Modality fusion, … |

(a) The original knowledge mastery state of employees.

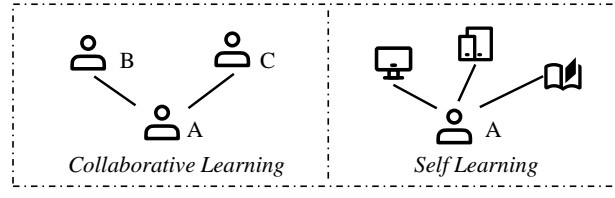

(b) The hybrid learning of employee A.

Python, PyTorch, Recommendation, …

↓

Python, PyTorch, Recommendation, API design, Modality fusion, Image recognition, Multimodal recommendation, …

(c) The knowledge development of employee A.

**Figure 1: An example of the knowledge development process of employee A with hybrid knowledge learning.**

one of the pivotal tasks in online KMS, tracking the knowledge development process of employees benefits proactively understanding their knowledge state and planning their future knowledge learning [5] for promoting career progress and stability.

During the past decades, researchers have devoted large efforts to predicting the knowledge development of students in the education field [26, 45]. However, these methods are not suitable for enterprise scenarios, since they ignore collaborative networks and knowledge flow to support quantitative and personalized knowledge development prediction for employees. Nowadays, enterprises are undergoing the shift to data-driven knowledge management, making management styles informative and intelligent. In this process, large-scale talent data have been significantly accumulated in online platforms, which implies the patterns of employee knowledge development and provides an unparalleled opportunity for achieving effective knowledge development prediction.

Indeed, knowledge development is a complicated process that involves the hybrid knowledge learning of employees including collaborative learning and self learning [23, 24]. Figure 1 shows an example of such a hybrid learning and knowledge development process. Specifically, Figure 1(a) shows the original knowledge mastery states of employee A, B, and C. After the hybrid learning in Figure 1(b), employee A acquires several new knowledge in Figure 1(c), which may come from different kinds of learning patterns. To be specific, "API design" and "Modality fusion" are more likely to be results of knowledge flow from collaborator B and C respectively,

while others probably depend on self learning. Apart from the hybrid learning process, knowledge development can have inherent rules. On the one hand, co-occurrence relationships exist between job knowledge, because work generally requires employees to learn multiple specific knowledge simultaneously. For example, a multimodal recommendation project requires employee A to know both "modality fusion" and "multimodal recommendation". On the other hand, knowledge is often learned from the shallower to the deeper. For employee A, "recommendation" is prerequisite knowledge for "multimodal recommendation". Motivated by this employee knowledge learning process, we intend to model and predict the knowledge development of employees from a hybrid learning perspective, with considerations of knowledge development rules.

However, it is non-trivial to model this hybrid learning for knowledge development prediction. First, job knowledge development is jointly influenced by multiple learning patterns and multiple collaborators [16]. These complicated situations make it difficult for the model to seize the pathways to job knowledge acquisition. Second, since the personal traits of employees (e.g., learning ability) may impact the way they propagate and learn knowledge, collaboration does not always bring job knowledge flow. Thus, it is difficult to model employees' ability to transfer and receive concrete knowledge and how they influence knowledge flow in collaboration. Third, leveraging the inherent rules of knowledge development requires the model to capture the relationships between knowledge. However, traditional statistical methods suffer from severe noise and are hard to estimate these relationships [7, 31].

To conquer these challenges, we propose a Collaboration-Aware Hybrid Learning approach (CAHL) to automatically predict the future knowledge acquisition of employees for knowledge development prediction. In CAHL, we first specially design a Job Knowledge Embedding (JKE) module to fully harness the inherent rules of job knowledge development. In particular, an association prompt attention mechanism is developed to capture the co-occurrence and prerequisite relationships between knowledge. Then, we aggregate the features of mastering knowledge and work collaborators for employee representations in another Employee Embedding (EE) module. Moreover, we propose a Hybrid Learning Simulation (HLS) module to predict future-acquired job knowledge, which models employee knowledge development via simulating collaborative and self learning patterns in a hybrid view. Specially, we model the collaborative learning process in terms of knowledge flow. In this part, the outflow and inflow score functions are invented to model employees' ability to transfer and receive concrete knowledge. For modeling the self learning process, a personalized knowledge acquisition score function is devised based on employee profiles. Finally, extensive experiments conducted on a real-world dataset clearly validate the effectiveness of CAHL.

## 2 RELATED WORK

In this section, we introduce the related work on knowledge development prediction and graph representation learning.

### 2.1 Knowledge Development Prediction

Existing studies on knowledge development prediction mainly concentrate on the knowledge state of students in the education area [3]. Early arts are proposed to utilize Bayesian methods to assume that student knowledge is represented as a set of binary variables [3, 32, 45]. Recently, researchers have started to leverage deep learning techniques to update the knowledge states of students with side information, such as the contents of exercises [2, 26], knowledge point graphs [31], and knowledge-exercise relationships [15, 19, 27]. Although knowledge development prediction for students has been performed through the above methods, they do not consider the knowledge development of employees based on their work collaborations and ecological situation in workplaces. Therefore, they cannot be directly applied in enterprise scenarios. In contrast to previous methods, we focus on knowledge development prediction for employees with new challenges in this paper.

### 2.2 Graph Representation Learning

Graph representation learning is proposed to embed nodes in the graph into a low-dimensional space for downstream applications. Early studies on graph representation can be roughly grouped into matrix factorization methods [1, 34] and random walk methods [8, 33]. In recent years, Graph Neural Networks (GNNs) have emerged with the rise of deep learning [9, 22, 36]. They primarily obey the message passing paradigm to aggregate the information of neighboring nodes. However, the real-world graph usually comes with multi-types of nodes and edges, which boosts the research of Heterogeneous Graph Neural Networks (HGNNs). Some methods utilize one-hop neighbors to aggregate such as HGAT [12], HGT [13], and HINormer [29]. They assign heterogeneous attention to either nodes or edges in original graphs. Other HGNNs such as HAN [39], HGSL [46], and HPN [17] exploit meta-paths to generate a new graph to learn the representations.

## 3 PRELIMINARY

In this section, we first describe the real-world data used in this paper. Then, we formally illustrate the problem definition of knowledge development prediction. Afterwards, we explain how the employee-knowledge graph is constructed for knowledge development prediction task. Table 1 shows the mathematical symbols.

### 3.1 Data Description

We used a set of in-firm data provided by a high-tech company, which was automatically collected through an online KMS, spanning the time period of 2018 to 2020. Note that, all of the sensitive information in the data has been removed or anonymized for privacy protection. It contains the profile data including basic information about employees, knowledge data including knowledge mastered by employees from annual talent reviews, and collaboration data including collaborators and collaboration times according to project records. More details are contained in Appendix. To verify the feasibility of knowledge development prediction task, we analyze the distributions of future-acquired knowledge ratio and latent flowing knowledge ratio. The statistical results of 2019 and 2020 are visualized in Figure 2. The future-acquired knowledge ratio of an employee is the ratio of their future-acquired knowledge to all their mastered knowledge. The average future-acquired knowledge ratios are 0.396 in 2019 and 0.372 in 2020. The two figures located above suggest that almost all employees in the data have

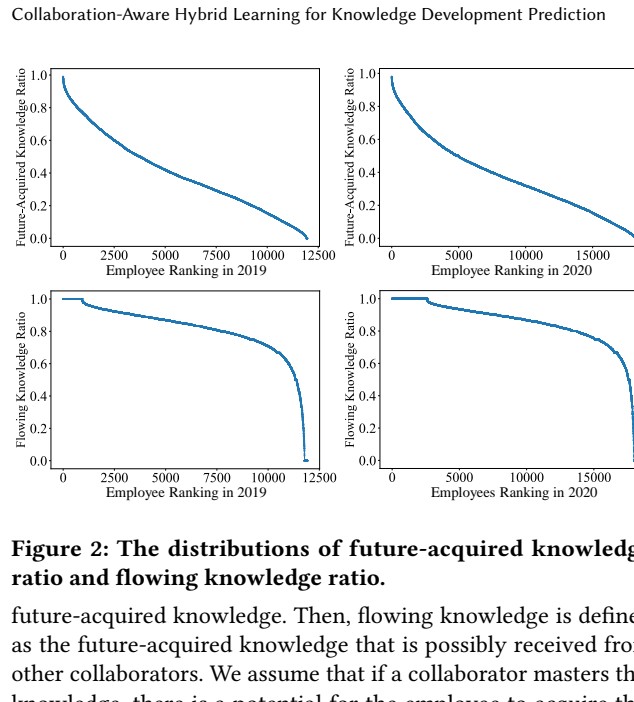

**Figure 2: The distributions of future-acquired knowledge ratio and flowing knowledge ratio.**

future-acquired knowledge. Then, flowing knowledge is defined as the future-acquired knowledge that is possibly received from other collaborators. We assume that if a collaborator masters this knowledge, there is a potential for the employee to acquire this knowledge through knowledge flow. The flowing knowledge ratio of an employee is the ratio of flowing knowledge to all their future-acquired knowledge. As shown in the figures located below, we can observe that, for the great majority of employees, flowing knowledge occupies more than half of their future-acquired knowledge. This indicates that the acquisition of new knowledge is highly relevant to knowledge flow in work collaborations.

### 3.2 Problem Definition

For an employee $e_i$, given an employee profile feature $u_i$, a set of knowledge mastery state $\mathcal{K}(e_i)$, and a set of collaboration records $C(e_i)$, knowledge development prediction targets to predict the future-acquired knowledge set $\mathcal{K}'(e_i)$. Here, $\mathcal{K}(e_i)$ is the set of knowledge that $e_i$ currently masters. $C(e_i) = \{(e_j, c_{ij}) | e_j \in \mathcal{H}(e_i)\}$ is the set of collaboration records of $e_i$, where $\mathcal{H}(e_i)$ is the set of collaborators with $e_i$, and $c_{ij}$ is the frequency of collaboration between $e_i$ and $e_j$.

### 3.3 Employee-Knowledge Graph Construction

To better model the complicated relationships between knowledge and employees, we construct an employee-knowledge graph with knowledge mastery states and collaboration records. Specifically, the employee-knowledge graph is defined as $\mathcal{G} = (\mathcal{V}, \mathcal{E})$, where $\mathcal{V} = \mathcal{V}_e \cup \mathcal{V}_k$ and $\mathcal{E} = \mathcal{E}_{ee} \cup \mathcal{E}_{ek} \cup \mathcal{E}_{kk}$ denote the set of nodes and edges, respectively. In particular, we use $\mathcal{V}_e$ and $\mathcal{V}_k$ to denote employee nodes and job knowledge nodes. Each edge in $\mathcal{E}_{ee}$ indicates that employees collaborated with each other. For example, if $(e_j, c_{ij}) \in C(e_i)$, there exists an edge $(e_i, e_j) \in \mathcal{E}_{ee}$. Each edge in $\mathcal{E}_{ek}$ indicates that the employee has mastered the knowledge. Analogously, if $k_j \in \mathcal{K}(e_i)$, there exists an edge $(e_i, k_j) \in \mathcal{E}_{ek}$. Besides, $\mathcal{E}_{kk} = \mathcal{E}_{kk}^c \cup \mathcal{E}_{kk}^p \cup \mathcal{E}_{kk}^m$, where $\mathcal{E}_{kk}^c$ indicates the co-occurrence relationship between knowledge, $\mathcal{E}_{kk}^p$ indicates the prerequisite relationship between knowledge, $\mathcal{E}_{kk}^m$ indicates the association prompt relationship between knowledge. According to the method

**Table 1: Mathematical symbols in the preliminary.**

| Symbol | Description |
|---|---|
| $e_i$ | The employee $i$; |
| $k_i$ | The knowledge $i$; |
| $u_i$ | The profile feature of $e_i$; |
| $\mathcal{K}(e_i)$ | The set of knowledge mastery state of $e_i$; |
| $\mathcal{K}'(e_i)$ | The set of future-acquired knowledge of $e_i$; |
| $C(e_i)$ | The set of collaboration records of $e_i$; |
| $\mathcal{H}(e_i)$ | The set of collaborators of $e_i$; |
| $c_{ij}$ | The frequency of collaboration between $e_i$ and $e_j$; |
| $\mathcal{G}$ | The employee-knowledge graph; |
| $\mathcal{V}$ | The set of nodes in $\mathcal{G}$; |
| $\mathcal{V}_e$ | The set of employee nodes in $\mathcal{G}$; |
| $\mathcal{V}_k$ | The set of job knowledge nodes in $\mathcal{G}$; |
| $\mathcal{E}$ | The set of edges in $\mathcal{G}$; |
| $\mathcal{E}_{ee}$ | The set of collaboration edges in $\mathcal{G}$; |
| $\mathcal{E}_{ek}$ | The set of knowledge mastery edges in $\mathcal{G}$; |
| $\mathcal{E}_{kk}$ | The set of knowledge relationship edges in $\mathcal{G}$; |
| $\mathcal{E}_{kk}^c$ | The set of co-occurrence relationship edges in $\mathcal{G}$; |
| $\mathcal{E}_{kk}^p$ | The set of prerequisite relationship edges in $\mathcal{G}$; |
| $\mathcal{E}_{kk}^m$ | The set of association prompt relationship edges in $\mathcal{G}$. |

in [7], we judge that knowledge has a co-occurrence relationship when they frequently occur in the same knowledge mastery state, and knowledge has a prerequisite relationship when one knowledge occurs in the set of future-acquired knowledge and the other knowledge occurs in the set of knowledge mastery state frequently. The association prompt relationship is extracted based on the meta-path in the form of $k_i \xrightarrow{\mathcal{E}_{ek}} e_m \xrightarrow{\mathcal{E}_{ee}} e_n \xrightarrow{\mathcal{E}_{ek}} k_j$, where $k_i, k_j \in \mathcal{V}_k$ and $e_m, e_n \in \mathcal{V}_e$.

## 4 METHODOLOGY

In this section, we introduce the technical details of our proposed CAHL. As illustrated in Figure 3, CAHL consists of three main modules, i.e., Job Knowledge Embedding (JKE) to capture the co-occurrence and prerequisite relationships between knowledge via an association prompt attention mechanism for job knowledge representations, Employee Embedding (EE) to aggregate the features of mastering knowledge and work collaborators for employee representations, and Hybrid Learning Simulation (HLS) to model the process of collaborative learning and self learning simultaneously to predict knowledge development for employees.

### 4.1 Job Knowledge Embedding

In this module, to fully harness the inherent rules of job knowledge development and learn better job knowledge representations, we design an association prompt attention mechanism to capture the co-occurrence relationships and prerequisite relationships between job knowledge.

*4.1.1 Co-Occurrence Relationship.* Co-occurrence knowledge refers to the simultaneous occurrence of two job knowledge from the same job demand. In Figure 1, "modality fusion" and "multimodal recommendation" have a co-occurrence relationship. In this part, we only focus on the sub-graph $\mathcal{G}_{kk}^c = (\mathcal{V}_k, \mathcal{E}_{kk}^c)$ presenting the

**Figure 3: The framework overview of a Collaboration-Aware Hybrid Learning approach (CAHL), which comprises three main components, i.e., Job Knowledge Embedding (JKE), Employee Embedding (EE), and Hybrid Learning Simulation (HLS).**

co-occurrence relationships between knowledge and exploit Graph-SAGE [9] to represent the job knowledge. The randomly initialized embeddings for all job knowledge nodes are provided as the input. For example, the input $h_{c_i}^0$ for job knowledge $k_i$ is the random initialized embedding $k_i$. For each layer, all job knowledge aggregates the features of nodes in their immediate neighborhood with co-occurrence frequency as weight. Next, the hidden and aggregated feature vectors are concatenated to generate a co-occurrence feature with a transformation matrix and activation function. In $l$-th layer, co-occurrence feature vector of $k_i$ are formulated as follows:

$$h_{N_{c_i}}^{(l)} = Agg(\{a_{ij}^c h_{c_j}^{(l-1)}, \forall (k_i, k_j) \in \mathcal{E}_{kk}^c\}), \quad (1)$$

$$h_{c_i}^{(l)} = ReLU(\mathbf{W}_c^{(l)}(h_{c_i}^{(l-1)} \oplus h_{N_{c_i}}^{(l)})), \quad (2)$$

where $a_{ij}^c$ is the normalized co-occurrence frequency of $k_i$ and $k_j$, $h_{c_i}^{(l)}$ and $h_{N_{c_i}}^{(l)}$ are the hidden feature vector and aggregated feature vector of $k_i$ in $l$-th layer, $\oplus$ is the concatenation operation, $ReLU(\cdot)$ is the activation function, and $Agg(\cdot)$ is the mean aggregator. Especially, the activation function is removed in the last layer. After $l_R$ layers, we obtain the final co-occurrence feature embedding $h_{c_i}$.

*4.1.2 Prerequisite Relationship.* Prerequisite refers to the relationship that low-level knowledge is required as a prior condition for mastering some high-level knowledge. As the case in Figure 1, "recommendation" is prerequisite knowledge for "multimodal recommendation". Analogously, it is possible that the knowledge possessed by the employee in the previous state is prerequisite knowledge for some knowledge in the current state. The prerequisite relationship of job knowledge is also extracted through statistics. Then, the same GraphSAGE structure is applied on the sub-graph $\mathcal{G}_{kk}^p = (\mathcal{V}_k, \mathcal{E}_{kk}^p)$ to learn prerequisite features for job knowledge

representations. For job knowledge $k_i$, we adopt the random initialized embedding $k_i$ as the input $h_{p_i}^0$. Then, the aggregation operation is in the following:

$$h_{N_{p_i}}^{(l)} = Agg(\{a_{ij}^p h_{p_j}^{(l-1)}, \forall (k_i, k_j) \in \mathcal{E}_{kk}^p\}), \quad (3)$$

$$h_{p_i}^{(l)} = ReLU(\mathbf{W}_p^{(l)}(h_{p_i}^{(l-1)} \oplus h_{N_{p_i}}^{(l)})), \quad (4)$$

where $a_{ij}^p$ is the normalized prerequisite frequency of $k_i$ and $k_j$, and $h_{p_i}^{(l)}, h_{N_{p_i}}^{(l)}$ are the hidden feature vector and aggregated feature vector of $k_i$ in $l$-th layer. Finally, the prerequisite feature embedding $h_{p_i}$ is generated without the activation function in $l_R$-th layer.

*4.1.3 Association Prompt Attention.* Intuitively, the relationship between the mastering knowledge of employees and collaborators can prompt the co-occurrence and prerequisite relationships learning for job knowledge. As mentioned before, the association prompt relationship is extracted based on the meta-path in the form of $k_i \xrightarrow{\mathcal{E}_{ek}} e_m \xrightarrow{\mathcal{E}_{ee}} e_n \xrightarrow{\mathcal{E}_{ek}} k_j$, where $k_i, k_j \in \mathcal{V}_k$ and $e_m, e_n \in \mathcal{V}_e$. Thus, we first leverage the sub-graph $\mathcal{G}_{kk}^m = (\mathcal{V}_k, \mathcal{E}_{kk}^m)$ and use $k_i$ as the input $h_{m_i}^0$ to aggregate the neighboring features as follows:

$$h_{N_{m_i}}^{(l)} = Agg(\{h_{m_j}^{(l-1)}, \forall (k_i, k_j) \in \mathcal{E}_{kk}^m\}), \quad (5)$$

$$h_{m_i}^{(l)} = ReLU(\mathbf{W}_m^{(l)}(h_{m_i}^{(l-1)} \oplus h_{N_{m_i}}^{(l)})), \quad (6)$$

where $h_{m_i}^{(l)}$ and $h_{N_{m_i}}^{(l)}$ are the hidden feature vector and aggregated feature vector of $k_i$ in $l$-th layer. In this way, we generate the association prompt feature embedding $h_{m_i}$ without the activation function in the last $l_R$-th layer. To further enhance knowledge relationships learning, we invent an association prompt attention mechanism to promote co-occurrence and prerequisite relationship

learning. The attention weights of co-occurrence and prerequisite features are calculated as follows:

$$(\beta_c, \beta_p) = Att(\mathbf{H}_c, \mathbf{H}_p, \mathbf{H}_m), \tag{7}$$

where $\beta_c$, $\beta_p$ are the attention weights of co-occurrence and prerequisite features, $Att(\cdot)$ denotes the association prompt attention mechanism, and $\mathbf{H}_c$, $\mathbf{H}_p$, and $\mathbf{H}_m$ are co-occurrence, prerequisite, and association prompt features of all job knowledge. For each job knowledge, we measure the similarity between the projected feature embedding and the association prompt embedding as the confidence of the co-occurrence and prerequisite features. The confidence scores of all co-occurrence and prerequisite embeddings are averaged as their attention weights:

$$w_\Phi = \frac{1}{|\mathcal{V}_{\mathcal{K}}|} \sum_{k_i \in \mathcal{V}_{\mathcal{K}}} \boldsymbol{h}_{m_i}^T \tanh(\mathbf{W}_{Att}\boldsymbol{h}_{\Phi_i} + \mathbf{b}_{Att}), \tag{8}$$

where $\Phi = c$ or $p$, $\mathcal{V}_{\mathcal{K}}$ is the set of all job knowledge nodes. We normalize the attention weights by a softmax function:

$$\beta_c = \frac{\exp(w_c)}{\exp(w_c) + \exp(w_p)}, \beta_p = \frac{\exp(w_p)}{\exp(w_c) + \exp(w_p)}. \tag{9}$$

After that, we fuse the co-occurrence and prerequisite features to obtain the final job knowledge embeddings with the attention weights as follows:

$$\boldsymbol{h}_i = \beta_c \cdot \boldsymbol{h}_{c_i} + \beta_p \cdot \boldsymbol{h}_{p_i}, \tag{10}$$

where $\boldsymbol{h}_i$ is the final embedding of job knowledge $k_i$.

## 4.2 Employee Embedding

In this module, for the purpose of considering employee characteristics in the behavior of job knowledge learning, we aggregate the features of mastering knowledge and work collaborators to represent the employees.

*4.2.1 Knowledge Mastery State.* The state of knowledge mastery portrays the current knowledge structure of employees, which allows inferring more accessible knowledge based on already acquired knowledge. In this part, we only focus on the sub-graph $\mathcal{G}_{ek} = (\mathcal{V}, \mathcal{E}_{ek})$ to learn the knowledge mastery feature of employees. First, we project the profile feature $\boldsymbol{u}_i$ of employee $e_i$ for reducing the dimension:

$$\boldsymbol{e}_i = \mathbf{W}_u \boldsymbol{u}_i + \mathbf{b}_u, \tag{11}$$

where $\boldsymbol{e}_i$ is the projected profile feature of employee $e_i$. We use $\boldsymbol{e}_i$ as the input $\boldsymbol{s}_{k_i}^0$ and concatenate $\boldsymbol{h}_j$ and $\boldsymbol{k}_j$ for the aggregation. For the employee $e_i$, we aggregate all the transformed features of mastering knowledge and employ multi-layer linear transformation to update the node features:

$$\boldsymbol{q}_i^{\mathcal{E}_{ek}} = Agg(\{\mathbf{W}_q^{\mathcal{E}_{ek}}(\boldsymbol{h}_j \oplus \boldsymbol{k}_j) + \mathbf{b}_q^{\mathcal{E}_{ek}}, \forall(e_i, k_j) \in \mathcal{E}_{ek}\}), \tag{12}$$

$$\boldsymbol{s}_{k_i}^{(l)} = ReLU(\mathbf{W}_{ek}^{(l)}(\boldsymbol{q}_i^{\mathcal{E}_{ek}} \oplus \boldsymbol{s}_{k_i}^{(l-1)})), \tag{13}$$

where $\boldsymbol{q}_i^{\mathcal{E}_{ek}}$ is the aggregated feature vector of $e_i$, and $\boldsymbol{s}_{k_i}^{(l)}$ is the hidden feature vector of $e_i$ in $l$-th layer. After $l_S$-layer transformation, $\boldsymbol{s}_{k_i}$ is obtained without the activation function in the last layer. Then, we project the knowledge mastery features into the same space for later fusion:

$$\boldsymbol{s}_i^{\mathcal{G}_{ek}} = \mathbf{W}_{\mathcal{G}_{ek}} \boldsymbol{s}_{k_i} + \mathbf{b}_{\mathcal{G}_{ek}}, \tag{14}$$

where $\boldsymbol{s}_i^{\mathcal{G}_{ek}}$ is the final knowledge mastery feature of employee $e_i$ in the sub-graph $\mathcal{G}_{ek}$.

*4.2.2 Collaboration State.* Typically, collaboration records of employees indicate the collaborators and collaboration frequency. Here, we intend to consider the features of work collaborators for better employee representation. In this part, we concentrate on the sub-graph $\mathcal{G}_{ee} = (\mathcal{V}_e, \mathcal{E}_{ee})$ about collaboration state to learn the collaboration features of employees. To begin with, the projected profile feature of employee $\boldsymbol{e}_i$ is input as $\boldsymbol{s}_{c_i}^0$. Then, we concatenate the final knowledge mastery feature and profile feature of employee and make a transformation for them. With regard to the employee $e_i$, all transformed features are aggregated to generate the hidden features, and multi-layer linear transformation is adopted to update the node features:

$$\boldsymbol{q}_i^{\mathcal{E}_{ee}} = Agg(\{c_{ij}\mathbf{W}_q^{\mathcal{E}_{ee}}(\boldsymbol{s}_j^{\mathcal{G}_{ek}} \oplus \boldsymbol{e}_j) + \mathbf{b}_q^{\mathcal{E}_{ee}}, \forall(e_i, e_j) \in \mathcal{E}_{ee}\}), \tag{15}$$

$$\boldsymbol{s}_{c_i}^{(l)} = ReLU(\mathbf{W}_{ee}^{(l)}(\boldsymbol{q}_i^{\mathcal{E}_{ee}} \oplus \boldsymbol{s}_{c_i}^{(l-1)})), \tag{16}$$

where $\boldsymbol{q}_i^{\mathcal{E}_{ee}}$ denotes the aggregated feature vector of $e_i$, $c_{ij}$ denotes the normalized collaboration frequency between $e_i$ and $e_j$, and $\boldsymbol{s}_{c_i}^{(l)}$ denotes the hidden feature vector of $e_i$ in $l$-th layer. In $l_S$-th layer, $\boldsymbol{s}_{k_i}$ is generated without the activation. Identically, we project the collaboration feature into the same space for later fusion:

$$\boldsymbol{s}_i^{\mathcal{G}_{ee}} = \mathbf{W}_{\mathcal{G}_{ee}} \boldsymbol{s}_{c_i} + \mathbf{b}_{\mathcal{G}_{ee}}, \tag{17}$$

where $\boldsymbol{s}_i^{\mathcal{G}_{ee}}$ is the final collaboration feature vector of employee $e_i$ in the sub-graph $\mathcal{G}_{ee}$.

For the purpose of fusing knowledge mastery and collaboration features, we concatenate two types of features and make another transformation for employee representation:

$$\boldsymbol{s}_i = LeakyReLU(\mathbf{W}_F(\boldsymbol{s}_i^{\mathcal{G}_{ek}} \oplus \boldsymbol{s}_i^{\mathcal{G}_{ee}}) + \mathbf{b}_F), \tag{18}$$

where $\boldsymbol{s}_i$ is the final embedding of employee $e_i$, $LeakyRuLU(\cdot)$ is the activation function.

## 4.3 Hybrid Learning Simulation

As we all know, job knowledge development results from the learning of knowledge by employees. In the real world, job knowledge learning is in the form of hybrid learning including collaborative learning and self learning. Inspired by this situation, we intend to model the process of employee knowledge development via the hybrid learning simulation.

*4.3.1 Collaborative Learning.* Generally, employees will learn partial job knowledge that their collaborators possess during project collaborations, which is a form of knowledge flow [6]. Furthermore, a successful knowledge flow is influenced by the ability of employees to transfer and receive concrete job knowledge. Therefore, we model the collaborative learning process in terms of knowledge flow. Specifically, the outflow and inflow score functions are invented to quantify an employees' ability to transfer and receive concrete knowledge. Collaborators can only transfer the knowledge they have acquired. Hence, we design the mastery score function by the dot product operation as follows:

$$\mathcal{F}_M(e_i, k_j) = \begin{cases} \boldsymbol{h}_j^T(\mathbf{W}_M \boldsymbol{s}_i + \mathbf{b}_M), \forall(e_i, k_j) \in \mathcal{E}_{ek}, \\ 0, \forall(e_i, k_j) \notin \mathcal{E}_{ek}. \end{cases} \tag{19}$$

$$\mathcal{M}(e_i, \mathcal{V}_k) = (\mathcal{F}_M(e_i, k_1), \mathcal{F}_M(e_i, k_2), ...), \tag{20}$$

where $\mathcal{F}_M(e_i, k_j)$ indicates how well employee $e_i$ has mastered the knowledge $k_j$ and $\mathcal{M}(e_i, \mathcal{V}_k)$ is the $|\mathcal{V}_k|$-dimension mastery score vector for all job knowledge. With the score of knowledge mastery, we further design the outflow score function with collaborator feature and collaboration frequency as follows:

$$\begin{aligned} O(e_x, e_i, \mathcal{V}_k) &= c_{xi}(\mathbf{W}_O \mathcal{M}(e_x, \mathcal{V}_k) + \mathbf{b}_O) \\ &= (\mathcal{F}_O(e_x, e_i, k_1), \mathcal{F}_O(e_x, e_i, k_2), ...), \end{aligned} \tag{21}$$

where $\mathcal{F}_O(e_x, e_i, k_j)$ is the outflow score function to indicate the probability that collaborator $e_x$ transfers the job knowledge $k_j$ to employee $e_i$, $c_{xi}$ is the normalized collaboration frequency between $e_x$ and $e_i$, and $O(e_x, e_i, \mathcal{V}_k)$ is the $|\mathcal{V}_k|$-dimension outflow score vector for all job knowledge. Usually, the contribution of knowledge acquisition may come from multiple collaborators in multiple collaborations. Considering the cumulative contribution of multiple collaborators with employee features, we specially design a GRU-cell structure to quantify an employee's ability to receive knowledge for state update. The employee embedding $s_i$ is the input and the cumulative contribution is treated as the hidden vector of the GRU-cell as follows:

$$\begin{aligned} \mathcal{I}(e_i, \mathcal{V}_k) &= \text{GRU}_{\text{cell}}(s_i, \sum_{(e_x, e_i) \in \mathcal{E}_{ee}} O(e_x, e_i, \mathcal{V}_k)) \\ &= (\mathcal{F}_I(e_i, k_1), \mathcal{F}_I(e_i, k_2), ...), \end{aligned} \tag{22}$$

where $\mathcal{F}_I(e_i, k_j)$ denotes the inflow score function for employee $e_i$ to receive knowledge $k_j$, $\mathcal{I}(e_i, \mathcal{V}_k)$ denotes the $|\mathcal{V}_k|$-dimension inflow score vector for all knowledge, and $\text{GRU}_{\text{cell}}(\cdot)$ denotes the GRU-cell structure. Furthermore, we set a constraint loss for knowledge flow by viewing knowledge that cannot be transferred by collaborators as negative samples. The mastery score function is designed for negative samples in the following:

$$\mathcal{F}'_M(e_i, k_j) = \begin{cases} \boldsymbol{h}_j^T(\mathbf{W}_M s_i + \mathbf{b}_M), \forall (e_i, k_j) \notin \mathcal{E}_{ek}, \\ 0, \forall (e_i, k_j) \in \mathcal{E}_{ek}, \end{cases} \tag{23}$$

$$\mathcal{M}'(e_i, \mathcal{V}_k) = (\mathcal{F}'_M(e_i, k_1), \mathcal{F}'_M(e_i, k_2), ...), \tag{24}$$

where $\mathcal{F}'_M(e_i, k_j)$ and $\mathcal{M}'(e_i, \mathcal{V}_k)$ are the functions for negative samples. Correspondingly, the outflow score vector for negative samples is defined as $O'(e_x, e_i, \mathcal{V}_k)$ in the same manner.

$$\begin{aligned} O'(e_x, e_i, \mathcal{V}_k) &= c_{xi}(\mathbf{W}_O \mathcal{M}'(e_x, \mathcal{V}_k) + \mathbf{b}_O) \\ &= (\mathcal{F}'_O(e_x, e_i, k_1), \mathcal{F}'_O(e_x, e_i, k_2), ...), \end{aligned} \tag{25}$$

Then, we devise the constraint loss to contrast positive examples with negative examples for distinguishing knowledge flow in collaborations as follows:

$$\begin{aligned} \mathcal{L}_c = -\frac{1}{|\mathcal{V}_e|} \sum_{e_i \in \mathcal{V}_e} Softplus(ReLU(\sum_{(e_x, e_i) \in \mathcal{E}_{ee}} O(e_x, e_i, \mathcal{V}_k)) \\ - ReLU(\sum_{(e_x, e_i) \in \mathcal{E}_{ee}} O'(e_x, e_i, \mathcal{V}_k))), \end{aligned} \tag{26}$$

where $\mathcal{L}_c$ is the constraint loss for knowledge flow, and $Softplus(\cdot)$ is the activation function.

*4.3.2 Self Learning.* Employees sometimes need to independently self-learn knowledge to satisfy their fast-changing job requirements. Hence, it is vital to model the self learning process of employees for knowledge development prediction. For self learning simulation, we deem that employee characteristics influence knowledge acquisition. Then, a personalized knowledge acquisition function is developed according to the employee profiles as follows:

$$\mathcal{F}_S(e_i, k_j) = \boldsymbol{h}_j^T(\mathbf{W}_S e_i + \mathbf{b}_S), \tag{27}$$

where $\mathcal{F}_S(e_i, k_j)$ indicates the probability that employee $e_i$ acquires knowledge $k_j$ from self learning.

Finally, the sum of the inflow score function for collaborative learning and the personalized knowledge acquisition score function for self learning is the output to predict the probability of acquiring new job knowledge:

$$\mathcal{F}(e_i, k_j) = \mathcal{F}_I(e_i, k_j) + \mathcal{F}_S(e_i, k_j), \tag{28}$$

where $\mathcal{F}(e_i, k_j)$ is the output of our model.

## 4.4 Model Training

In training stage, we design the overall objective function with the constraint loss $\mathcal{L}_c$ to update the model parameters:

$$\begin{aligned} \mathcal{L} = -\frac{1}{n} \sum_i \sum_j (y_{ij} \log(\sigma(\mathcal{F}(e_i, k_j))) \\ + (1 - y_{ij}) \log(1 - \sigma(\mathcal{F}(e_i, k_j)))) + \mathcal{L}_c, \end{aligned} \tag{29}$$

where $\mathcal{L}$ is the overall objective function, $\sigma$ is the sigmoid function, and $y_{ij}$ is the indicator of whether employee $e_i$ will acquire job knowledge $k_j$.

## 5 EXPERIMENT

In this section, we conduct extensive experiments on the real-world dataset. We first describe the experimental setup and then present the experimental results as well as analyses.

## 5.1 Experimental Setup

*5.1.1 Dataset.* We use a real-world dataset sourced from an online KMS provided by a high-tech company, spanning the year 2018 to 2020. The dataset comprises 8,349 employees and 3,633 knowledge, including 2,492,205 knowledge mastery records, 353,002 collaboration records, 218,839 co-occurrence knowledge relationships, and 92,694 prerequisite knowledge relationships. Since knowledge data were collected from annual records, we select a one-year time interval to analyze changes in employee state. In training set, the input contains employee samples in 2018 and the output contains employee samples in 2019, respectively. In test set, the input contains employee samples in 2019 and the output contains employee samples in 2020, respectively.

*5.1.2 Evaluation Metrics.* In this task, $\mathcal{V}_k$ contains a large amount of knowledge. Besides, the quantity of future-acquired knowledge per sample is uncertain. Following similar tasks [18, 25], we focus on top predictions by varying k at 1, 3, 5, and 10 in precision at k (P@k) and normalized discounted cumulative gain at k (N@k) to evaluate the performance.

**Table 2: The performance of all methods for knowledge development prediction on the real-world dataset.**

| Model | P@1 | P@3 | P@5 | P@10 | N@3 | N@5 | N@10 |
|---|---|---|---|---|---|---|---|
| Popularity | 0.4405 | 0.4353 | 0.4212 | 0.3813 | 0.4366 | 0.4267 | 0.3989 |
| GCN | 0.5370±0.0014 | 0.4852±0.0002 | 0.4467±0.0004 | 0.3886±0.0003 | 0.4971±0.0003 | 0.4673±0.0001 | 0.4217±0.0002 |
| GraphSAGE | 0.5415±0.0052 | 0.4879±0.0013 | 0.4533±0.0009 | 0.3994±0.0008 | 0.5001±0.0020 | 0.4730±0.0014 | 0.4307±0.0011 |
| GAT | 0.5383±0.0022 | 0.4849±0.0009 | 0.4464±0.0007 | 0.3858±0.0032 | 0.4971±0.0008 | 0.4672±0.0008 | 0.4199±0.0021 |
| NeuMF | 0.5429±0.0053 | 0.4834±0.0053 | 0.4473±0.0045 | 0.3946±0.0021 | 0.4967±0.0054 | 0.4682±0.0048 | 0.4263±0.0030 |
| NGCF | 0.5479±0.0013 | 0.4861±0.0019 | 0.4498±0.0014 | 0.3977±0.0012 | 0.5000±0.0017 | 0.4711±0.0012 | 0.4295±0.0010 |
| LightGCN | 0.5534±0.0019 | 0.4937±0.0008 | 0.4582±0.0005 | 0.4032±0.0002 | 0.5074±0.0009 | 0.4792±0.0006 | 0.4357±0.0003 |
| UltraGCN | 0.5473±0.0031 | 0.4938±0.0036 | 0.4592±0.0029 | 0.4074±0.0042 | 0.5060±0.0034 | 0.4788±0.0029 | 0.4381±0.0033 |
| HAN | 0.5464±0.0025 | 0.4898±0.0032 | 0.4533±0.0033 | 0.3970±0.0039 | 0.5026±0.0030 | 0.4739±0.0031 | 0.4297±0.0036 |
| HGT | 0.5553±0.0019 | 0.4945±0.0031 | 0.4568±0.0062 | 0.4006±0.0084 | 0.5084±0.0022 | 0.4786±0.0043 | 0.4341±0.0063 |
| HeCo | 0.5418±0.0006 | 0.4860±0.0006 | 0.4456±0.0002 | 0.3801±0.0009 | 0.4987±0.0005 | 0.4674±0.0002 | 0.4164±0.0007 |
| HPN | 0.5555±0.0017 | 0.4954±0.0040 | 0.4620±0.0044 | 0.4051±0.0042 | 0.5082±0.0037 | 0.4826±0.0040 | 0.4394±0.0039 |
| DiffNet | 0.5555±0.0032 | 0.4961±0.0022 | 0.4615±0.0023 | 0.4109±0.0020 | 0.5094±0.0022 | 0.4820±0.0019 | 0.4418±0.0019 |
| SEPT | 0.5466±0.0025 | 0.4910±0.0010 | 0.4571±0.0014 | 0.4045±0.0016 | 0.5038±0.0011 | 0.4769±0.0009 | 0.4354±0.0009 |
| DESIGN | 0.5504±0.0035 | 0.4936±0.0030 | 0.4593±0.0029 | 0.4033±0.0036 | 0.5029±0.0014 | 0.4783±0.0026 | 0.4394±0.0019 |
| SI-GAN | 0.5529±0.0028 | 0.4958±0.0029 | 0.4617±0.0032 | 0.4078±0.0031 | 0.5075±0.0030 | 0.4819±0.0038 | 0.4389±0.0026 |
| **CAHL** | **0.5790±0.0010** | **0.5206±0.0013** | **0.4842±0.0004** | **0.4286±0.0013** | **0.5333±0.0011** | **0.5048±0.0003** | **0.4613±0.0010** |

*5.1.3 Compared Methods.* We compare CAHL with three groups of representative and competitive baselines. First, we select the most frequent knowledge by statistics as Popularity. Second, we compare methods modeling the knowledge mastery state of employees by homogeneous GNN and general recommendation methods. Third, we compare methods integrating knowledge mastery state and collaboration records by heterogeneous GNN and social recommendation methods. The baselines are introduced as follows:

- **Popularity**: a statistical method to select the most frequent knowledge in knowledge mastery states of all employees.
- **GCN** [22]: a homogeneous GNN method which designs a convolutional structure to aggregate neighboring features.
- **GraphSAGE** [9]: a homogeneous GNN method which learns to aggregate features from a local neighborhood.
- **GAT** [36]: a homogeneous GNN method which integrates a masked self-attention strategy to aggregate neighbor features with weights.
- **NeuMF** [11]: a general recommendation method which combines the linearity of matrix factorization and non-linearity of neural networks for modeling user-item latent structures.
- **NGCF** [38]: a general recommendation method which exploits the user-item graph structure and injects the collaborative signal into the graph embedding process.
- **LightGCN** [10]: a general recommendation method which simplifies the design of GCN and uses the weighted sum of the embeddings learned at all layers as the final embedding.
- **UltraGCN** [28]: a general recommendation method which skips infinite layers of message passing and resorts to approximate the limit of infinite-layer convolutions via a constraint loss.
- **HAN** [40]: a heterogeneous GNN method which proposes a novel heterogeneous GNN based on the hierarchical attention, including node-level and semantic-level attentions.
- **HGT** [14]: a heterogeneous GNN method which designs node- and edge-type dependent parameters to characterize the heterogeneous attention over each edge.

- **HeCo** [41]: a heterogeneous GNN method which captures local and high-order structures simultaneously, and then employs cross-view contrastive learning.
- **HPN** [17]: a heterogeneous GNN method which absorbs the local semantics of nodes and injects distinguishable semantics into node embedding in node-level aggregating and semantic fusion mechanism to fuse them.
- **DiffNet** [43]: a social recommendation method which designs a layer-wise influence propagation structure to model the latent user embeddings evolve in the social diffusion process.
- **SEPT** [44]: a social recommendation method which employs tri-training to mine self-supervision signals from other users with the multi-view encoding.
- **DESIGN** [35]: a social recommendation method which proposes a distillation enhanced social graph network by exploiting the knowledge distillation for interaction and social graphs.
- **SI-GAN** [37]: a social recommendation method which inherently fuses the adversarial learning enhanced social network feature and graph interaction feature.

## 5.2 Experimental Results

*5.2.1 Performance Comparison.* The experimental results of all methods for knowledge development prediction on the real-world dataset are illustrated in Table 2. From the overview, CAHL achieves the best performance on knowledge development prediction task. Specifically, we have the following observations. First, CAHL consistently outperforms all baselines in terms of all evaluation metrics. Besides, we test the statistical significance between CAHL and all baselines, and the results suggest that CAHL has significant improvements (*p-value* < 0.001) over them. Second, in most cases, methods that incorporate collaboration records generate better results than those that only consider knowledge mastery state. This indicates leveraging collaboration states to model the characteristics or behaviors of employees is important for knowledge development prediction in enterprise scenarios. Third, among the methods that

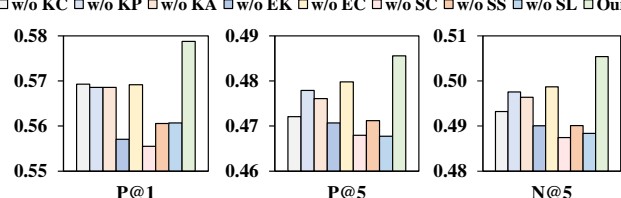

**Figure 4: The ablation study on the real-world dataset.**

incorporate collaboration records, CAHL obviously surpasses the others. This shows that CAHL can effectively model employees' knowledge flow with collaborative networks and explore multiple relationships of job knowledge. Last, although social recommendation methods focus on collaborative networks in the workplace, the improvement is not significant. This is because they mainly take employee similarity into account.

*5.2.2 Ablation Study.* To verify the contribution of each component in our proposed model, we design the three groups of variants as follows: 1) removing co-occurrence relationship, prerequisite relationship, and association prompt attention mechanism in JKE module, i.e., w/o KC, w/o KP, and w/o KA; 2) removing knowledge mastery state, and collaboration state in EE module, i.e., w/o EK, and w/o EC; 3) removing collaborative learning, self learning, and flow constraint loss in HLS module, i.e., w/o SC, w/o SS, and w/o SL. Figure 4 shows the performance of these variants, which demonstrate the effectiveness of each component in CAHL. Specifically, the third group of variants shows the worst performance in most cases, especially w/o SC. This indicates that our designed hybrid learning simulation can bring significant improvement, and collaborative learning plays a crucial role in this learning process. From the results of the second group, the performance of w/o EK drops obviously. This is because the knowledge mastery state of employees is core feature for the prediction. Besides, the first group of variants suggests the improvement of co-occurrence relationships increases as k increases, since it concerns multiple knowledge.

*5.2.3 Parameter Sensitivity.* In JKE module and EE module, we design the $l_R$- and $l_S$-layer aggregation for the update of knowledge and employee features, respectively. Therefore, we conduct experiments to study the impact of the number of $l_R$ and $l_S$ on the model performance. Here, we select the range of 1 to 5 as the number of layers for our experiments. Figure 5 shows the P@5 and N@5 scores with different numbers of network layers on the real-world dataset. The results of this study indicate that our proposed model performs at its best when the number of layers in both aggregation operations is set to two. With only one layer of aggregation, the model clearly does not take full advantage of the useful information from the neighbors. As the number of layers increases beyond two, the model performance decreases gradually. This is because an increase in the number of network layers leads to over-smoothing and introduces more noise information from multi-hop neighbors.

*5.2.4 Case Study.* Figure 6 shows two typical cases from the testing results. Due to the limited space, we only display the top-5 collaborators with the highest outflow scores. From these cases, we have the following observations. First, collaborative learning is influenced by the cumulative impact of multiple collaborators,

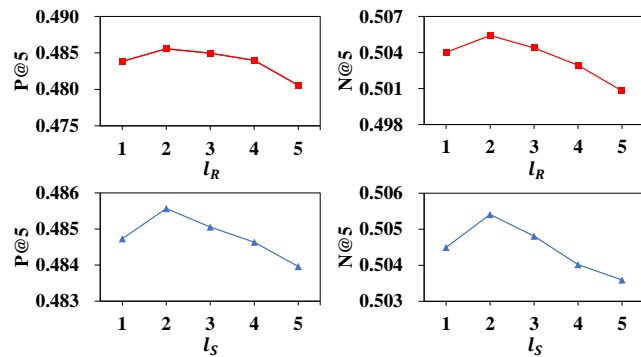

**Figure 5: The performance of CAHL with regard to different layer numbers, i.e., $l_R$ and $l_S$.**

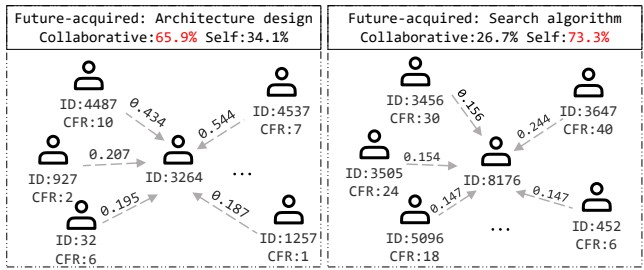

**Figure 6: The typical cases generated by CAHL. CFR denotes the ranking of collaborators in terms of frequency of collaboration with the employee. The weight of the edge between employees is the outflow score.**

and different collaborators exhibit varying abilities in transferring knowledge outward. In two cases, multiple collaborators transfer "Architecture design" to employee No.3264 and "Search algorithm" to employee No.8176, but the weights of the edges connecting them are different. Second, a higher frequency of collaboration does not necessarily result in a higher probability of transferring job knowledge. For example, in the right case, collaborators with the 40th most frequent collaboration have the highest probability of transferring knowledge. Third, knowledge acquired primarily through collaborative learning is more likely to be transferred by collaborators, and vice versa. The outflow score reaches 0.544 in the left case, while in the right case the highest score is only 0.244.

## 6 CONCLUSION

In this paper, we proposed a novel Collaboration-Aware Hybrid Learning approach (CAHL) for knowledge development prediction in workplaces. Specifically, we first learned the job knowledge representation by an association prompt attention mechanism to capture co-occurrence and prerequisite relationships between knowledge. This can fully harness inherent rules of knowledge development. Then, the features of mastering knowledge and work collaborators were aggregated to generate employee representations. Afterwards, we proposed to model the process of employee knowledge development via a hybrid learning simulation, including collaborative learning and self learning, to predict future-acquired job knowledge of employees. Finally, extensive experiments conducted on a real-world dataset clearly validated the effectiveness of CAHL.

# A APPENDIX

## A.1 Dataset Description

In this paper, a set of in-firm data were provided by a high-tech company and automatically collected through an online KMS, across a time span ranging from 2018 to 2020. For privacy protection, all of the sensitive information in the data has been removed or anonymized. The dataset we used includes three types of data in the following:

- **Profile data**: employee's profile vector. The basic information of employee is transformed into the embedding vector through an automated de-identification process in the KMS. It can be represented as ($employee\ ID$, $profile\ vector$).
- **Knowledge data**: employee's knowledge state. It can be represented as ($employee\ ID$, $year$, [$knowledge_1$, $knowledge_2$, ...]).
- **Collaboration data**: employee's collaboration record. It can be denoted as ($employee\ ID$, $year$, [($employee_1$, $times_1$), ($employee_2$, $times_2$), ...]).

It was easily and automatically collected through a knowledge management system without high costs involved. Talent review is a standard practice in knowledge management systems to label the knowledge state from self and peers. The determination made for mastery of knowledge is a part of the knowledge management system, and there is a knowledge tag library in the knowledge management system. In the annual talent review, each employee is given knowledge tags from the knowledge tag library by himself/herself and his/her colleagues to indicate his/her mastery of knowledge, which are then reviewed by his/her direct leader and HRBP. For example, employee A can select "Tensor Factorization" to label himself/herself, and employee B can select "Python" to label employee A.

## A.2 Difference with Education Scenario

As we mentioned before, the knowledge prediction methods for students' knowledge development have shown promising performance. However, they cannot be applied directly in enterprise scenarios, because they ignore the collaborative networks and knowledge flow in actual working environments. In the following, we explicitly state the main difference between these two scenarios:

- **The learning process of modeling.** In enterprise scenarios, project collaborations have a large impact on employee knowledge acquisition, which will cause the flow of knowledge in this process. Therefore, we mainly model the knowledge flow in collaborative networks. While in education scenarios, existing studies often focus on students' exercise records and model them, since students' knowledge acquisition mainly comes from doing exercises instead of collaborations.
- **The objects of modeling.** The objects modeled in enterprise scenarios are employees and knowledge, while in education scenarios, students, exercises, and knowledge are all modeled.

## A.3 Model Configuration

All weight matrices and random embeddings are initialized by the Xavier initializer with a uniform distribution. We set the dimension of initialized embeddings as 64. The dimension of input profile feature embeddings is 189. $l_R$ and $l_S$ are set as 2. The dimensions of co-occurrence and prerequisite hidden vectors are 512, and the dimension of other hidden vectors is 64. The dimension of all final feature embeddings is 64. The mini-batch method is adopted with the batch size of 256. We use Adam optimizer [21] with a learning rate of 0.001. The number of training epochs is 150. We repeat the experiments five times and report the average results. Our model is implemented with the deep learning framework PyTorch. The experiments are conducted on a server with two Intel(R) Xeon(R) Gold 6258R CPU @ 2.70GHz, and four NVIDIA GeForce RTX 2080 Ti GPUs. The code and data samples for CAHL are available at https://anonymous.4open.science/r/CAHL for reproducibility.

## A.4 Baseline Descriptions and Settings

We compare the proposed CAHL with several representative and state-of-the-art methods. The baseline settings are introduced in the following:

- **GCN** [22]: All weight matrices and random embeddings are initialized by the Xavier initializer with a uniform distribution. We set the dimension of initialized embeddings as 64. The dimension of hidden vectors is 64 and the layer number is 2. The mini-batch method is adopted with the batch size of 256. We use Adam optimizer with a learning rate of 0.001. The number of training epochs is 10.
- **GraphSAGE** [9]: All weight matrices and random embeddings are initialized by the Xavier initializer with a uniform distribution. We set the dimension of initialized embeddings as 64. The dimension of hidden vectors is 64 and the layer number is 2. The mini-batch method is adopted with the batch size of 256. We use Adam optimizer with a learning rate of 0.001. The number of training epochs is 50.
- **GAT** [36]: All weight matrices and random embeddings are initialized by the Xavier initializer with a uniform distribution. We set the dimension of initialized embeddings as 64. The dimension of hidden vectors is 64 and the layer number is 2. The number of attention heads is 2. The mini-batch method is adopted with the batch size of 256. We use Adam optimizer with a learning rate of 0.001. The number of training epochs is 50.
- **NeuMF** [11]: All weight matrices and random embeddings are initialized using the normal distribution with a mean of zero and standard deviation of 0.01. The embedding size is 64. The numbers of hidden units in MLP are [128, 64]. The mini-batch method is adopted with the batch size of 2048. We use Adam optimizer with a learning rate of 0.001. The number of training epochs is 10.
- **NGCF** [38]: All weight matrices and random embeddings are initialized by the Xavier initializer with a normal distribution. We set the dimension of initialized embeddings as 64. The dimension of hidden vectors is 64. The node dropout ratio is 0.0, and the message dropout ratio of 0.1. In the loss function, $\lambda = 1e^{-5}$. The mini-batch method is adopted with the batch size of 2048. We use Adam optimizer with a learning rate of 0.001. The number of training epochs is 50.
- **LightGCN** [10]: All weight matrices and random embeddings are initialized by the Xavier initializer with a uniform distribution. We set the dimension of initialized embeddings as 64. The

layer number is 2. In the loss function, $\lambda = 1e^{-5}$. The mini-batch method is adopted with the batch size of 2048. The number of training epochs is 50.

- **UltraGCN** [28]: All weight matrices and random embeddings are initialized by the Xavier initializer with a uniform distribution. We set the dimension of initialized embeddings as 64. We adopt L2 regularization with $1e^{-4}$ weight and set the learning rate to $1e^{-4}$, the batch size to 2048, the negative sampling ratio to 200, and the size of the neighbor set to 10. We use Adam optimizer with a learning rate of 0.001. The number of training epochs is 50.

- **HAN** [40]: All weight matrices and random embeddings are initialized by the Xavier initializer with a uniform distribution. We set the dimension of initialized embeddings as 64. The number of attention heads is 2. The dimension of the semantic-level attention vectors is 64. The mini-batch method is adopted with the batch size of 256. We use Adam optimizer with a learning rate of 0.001. The number of training epochs is 50.

- **HGT** [14]: All weight matrices and random embeddings are initialized by the Xavier initializer with a uniform distribution. We set the dimension of initialized embeddings as 64. We use 64 as the hidden dimension. The number of attention heads is 8. The layer number is set as 2. The mini-batch method is adopted with the batch size of 256. We use Adam optimizer with a learning rate of 0.001. The number of training epochs is 50.

- **HeCo** [41]: All weight matrices and random embeddings are initialized by the Xavier initializer with a uniform distribution. We set the dimension of initialized embeddings as 64. $\tau = 0.8$, $\lambda = 0.5$. The layer number is 1. The mini-batch method is adopted with the batch size of 256. We use Adam optimizer with a learning rate of 0.001. The number of training epochs is 100.

- **HPN** [17]: All weight matrices and random embeddings are initialized by the Xavier initializer with a uniform distribution. We set the dimension of initialized embeddings as 64. The layer number is 2 and $\gamma = 0.5$. The number of attention heads is 2. The dimension of the semantic-level attention vectors is 64. The mini-batch method is adopted with the batch size of 256. We use Adam optimizer with a learning rate of 0.001. The number of training epochs is 150.

- **DiffNet** [43]: All weight matrices and random embeddings are initialized by the Xavier initializer with a uniform distribution. We set the dimension of initialized embeddings as 64. The layer number is 2. In the loss function, $\lambda = 1e^{-5}$. The mini-batch method is adopted with the batch size of 2048. We use Adam optimizer with a learning rate of 0.001. The number of training epochs is 30.

- **SEPT** [44]: All weight matrices and random embeddings are initialized by the Xavier initializer with a uniform distribution. We set the dimension of initialized embeddings as 64. The layer number is 2. Besides, $\tau = 0.1$, $\rho = 0.3$, $K = 10$, $\beta = 1e^{-7}$, $\lambda = 1e^{-5}$. The mini-batch method is adopted with the batch size of 2048. We use Adam optimizer with a learning rate of 0.001. The number of training epochs is 50.

- **DESIGN** [35]: All weight matrices and random embeddings are initialized by the Xavier initializer with a uniform distribution. The embedding size is fixed to 64. We optimize the model using Adam optimizer with a learning rate of 0.001, where the batch size is fixed to 512. The number of training epochs is 50.

- **SI-GAN** [37]: All weight matrices and random embeddings are initialized by the Xavier initializer with a uniform distribution. The embedding size is fixed to 64. We optimize the model using Adam optimizer with a learning rate of 0.001, where the batch size is set to 512. In order to avoid over-fitting, dropout is adopted with a rate of 0.5. For the diffusion model, we set the layer number to 2 and 3 respectively. The number of training epochs is 50.

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
