# OpenReview forum: "Collaboration-Aware Hybrid Learning for Knowledge Development Prediction"
_ACM.org/TheWebConf/2024/Conference — TheWebConf24 Oral_

### Official Review · Reviewer_gTRD · 2023-11-08

**Novelty:** 5
**Technical Quality:** 4

**Review:**

The topic is interesting and relevant to the community and the need for better knowledge development prediction in enterprise environments is fairly well argued, as is the difference to the education scenario.

The authors present hybrid knowledge simulation methods and their performance in a real-world dataset. The detailed description of the graphs and embeddings are among the strengths of the paper. The experiment in with a real-world dataset and the comparison of the performance of different methods is another strength.

One of the weak points of the paper is that, despite the detail, the framework overview and the embeddings can be difficult to follow for people not working in this field. The same applies to the methodology, where the choice of methods to compare the performance of CAHL against is not sufficiently argued; have those been used in relevant research and in the literature reported in the background section? Also, when it comes to the description of the embeddings and their description, what where the possible approaches, what is the overall rationale, and how does that relate to the literature?

Further, the results section appears to focus on the performance of the proposed approach but the value of the paper would be stronger if there was also discussion on why the proposed approach is better in this experiment, what are the choices that need to be made in this or similar domains, and how organisations could proceed with the collection of data and knowledge prediction system. Further, even though enterprise environments are argued to be different to the educational ones it would strengthen the paper to have some discussion of how the performance of CAHL relates to relevant systems in the education domain.

More specifically:
- section 2.1: Even though it is argued that education systems are different in their knowledge prediction, what systems/methods have been used for knowledge prediction in that domain?
- section 3.1: The dataset description is high level (even in the appendix). Information on KMS used or sample records would help with the rigour and reproducibility of the paper.
- the explanations of the co-occurrence relationship, the prerequisite relationship, and the association prompt attention are hard to follow. There is an attempt of providing an example for the first one, in 4.1.1 but more example and explanation would be needed to ensure this work is approachable by the a wider part of the community.
- sections 4.2.1: This section is hard to follow and it would require some justification and explanation of the embeddings.
- section 5.1.3: The choice of methods to compare is not sufficiently justified in terms of relevance or use in the literature in similar contexts.
- section 6: The contribution of the paper is not as strong as it could be. I could provide a wider discussion of the contribution of this work especially with regard to other such systems (even if they are in education) and to organisations who employ knowledge management systems.

NOTE AFTER REBUTTAL
I acknowledge that I have read the rebuttals and I would like to thank the authors for their responses.

**Questions:**

- Which of the baselines in Table 2 have been used in relevant work in the domain of knowledge management development prediction?
- There is sufficient argumentation on the differences between the education scenario and this scenario but how does the performance of predication systems in education settings compare to the enterprise one or to other enterprise settings?

**Reviewer Confidence:**

2: The reviewer is willing to defend the evaluation, but it is likely that the reviewer did not understand parts of the paper

**Scope:**

2: The connection to the Web is incidental, e.g., use of Web data or API

---

### Official Review · Reviewer_rMnh · 2023-11-23

**Novelty:** 4
**Technical Quality:** 5

**Review:**

In this work the authors propose a framework  to tackle the problem of knowledge development prediction in the context of entreprises.  Their approach uses an association prompt attention mechanism to generate job knowledge representations by capturing the co-occurrence and prerequisite relationships between job knowledge. They additionally generate employee embeddings by aggregating features of acquiring this knowledge and from work collaborators. The whole model is trained as a hybrid learning simulation that includes collaborative learning and self-learning to make predictions about the future job knowledge of employees.

The motivation for this work is not novel on its own, as research on knowledge development has been around for some time, however its application on a business setting is an emerging and important aspect of it. The proposed model captures a variety of different factors that can influence knowledge propagation, while the graph modeling approach is a good representation when capturing correlations and co-dependences in the data. One part that wasn’t clear to me was the differences/overlaps between the existing student knowledge approaches and the current model - except the collaborative elements. That said, overall the presentation and flow of the paper was good, the methodology and experiments clearly presented. The authors had the opportunity to use a real world dataset for their experiments which aided their case. In addition, they compared their method to an extensive number of other approaches which makes for a solid experimentation setup. My one concern here would be that I wanted the authors to highlight the major differences of their work with the selected algorithms for experimentation, to better showcase their contribution (from a modeling prospective).

**Questions:**

- Could you summarize more clearly what the contributions of this work are, in comparison to the existing approaches? I read about the problems but I would expect something more concrete like: "these models don't deal with this, which we added", etc.

**Ethics Review Description:**

-

**Reviewer Confidence:**

2: The reviewer is willing to defend the evaluation, but it is likely that the reviewer did not understand parts of the paper

**Scope:**

4: The work is relevant to the Web and to the track, and is of broad interest to the community

---

### Official Review · Reviewer_xFZS · 2023-11-30

**Novelty:** 5
**Technical Quality:** 6

**Review:**

The paper presents a novel approach to predicting the knowledge/skills that someone will develop in the future. Based on user profile information, their knowledge areas, and collaboration information, the model generated prediction scores. The novelty of the approach is that it combines different ways that users might learn (individual effort, improvement through collaboration). A real dataset from industry is used to test the approach, and it indeed outperforms the competing approaches.

The strong points of the paper involve a well-thought-out model, that takes into consideration many aspects that can affect learning. While sometimes it seems that it is overly complex, the main idea is well communicated, and the notation is used consistently. The approach is compared against a number of other methods, and the significance of the findings is considered. The Appendix provides useful additional information about the data and the implementation. The paper provides the code as well, which is particularly useful for this case. The insights from the experimental evaluation are nicely presented, as well.

On the other hand, while the idea is more intuitive, the approach seems to be overly complicated, with a large number of parameters. There is no parameter tuning; all hyperparameters are set to fixed values. Most importantly, the paper requires a lot of specific prior knowledge from the potential readers, as some details/terms are not properly explained (e.g., future-acquired knowledge ratio, latent flowing knowledge ratio, association prompt relationship), limiting the audience that it would attract. Minor point: the colors in Fig. 4 are not easy to distinguish.

**Questions:**

What could such a model offer in a real scenario?

**Reviewer Confidence:**

3: The reviewer is confident but not certain that the evaluation is correct

**Scope:**

4: The work is relevant to the Web and to the track, and is of broad interest to the community

---

### Official Review · Reviewer_rGgT · 2023-11-30

**Novelty:** 6
**Technical Quality:** 6

**Review:**

This paper applies graph representation learning onto the problem of knowledge development prediction. This problem has real-world applications in knowledge management. The authors frame the problem as a social network problem which motivates the use of graph representation learning as a potential approach. I believe that the key contribution lies in the Hybrid Learning Simulation component where it seeks to model knowledge learnt via collaboration and knowledge learnt via self-agency.

Pros:
+ Large real-world dataset.
+ Extensive comparison across multiple other graph neural models.
+ Interesting case study on interpreting the edge-weights.

Cons:
- Only one dataset so we are unable to ascertain the generalizability, although it is understandable as such datasets are difficult to obtain.

I have read the rebuttal and the authors have answered my questions.

**Questions:**

Q1:  Will the full anonymized dataset be available upon publication?

Q2: Collaboration between individuals can change over time, such as when employees leave or are transferred. This might result in some information lost due as previous collaborators are blocked from rating someone they had worked with. How was that accounted for during the dataset preparation?

**Reviewer Confidence:**

3: The reviewer is confident but not certain that the evaluation is correct

**Scope:**

3: The work is somewhat relevant to the Web and to the track, and is of narrow interest to a sub-community

---

### Official Review · Reviewer_kRcC · 2023-11-30

**Novelty:** 5
**Technical Quality:** 6

**Review:**

This paper addresses the task of modeling hybrid learning for predicting job knowledge development. Traditional statistical methods struggle with noise and relationship estimation in knowledge development. To overcome these challenges, the authors propose a collaboration-aware hybrid learning approach, which incorporates a job knowledge embedding module to capture inherent rules, an employee embedding module for employee representations, and a hybrid learning simulation module for predicting future-acquired job knowledge. Experimental results on a real-world dataset show the effectiveness of he proposed approach in addressing these complexities.

Strengths
+ The method appears sound and technically convincing, with solid mathematical foundations
+ Overall well-documented paper, including appendix, which enables readers to understand the key messages and contributions
+ There is a large number of baselines covered in the experiments, making the comparison strong from this perspective

Limitations
- Experiments focus only on one dataset, limiting the generalizability of the proposed model, and such dataset would not be publicly available (only some samples are being shared within the appendix), which prevents to reproduce the proposed method and replicate the results. I invite the authors to consider to add experiments on at least an additional public dataset, regardless of the availability of the proposed one, to address both issues.
- The core of the methodology (Section 4) is very dense in terms of mathematical notation and formulas; in addition to this, the core ideas are often presented without a more conceptual introduction of each component that would make the paper more accessible to a broader audience, for instance that focusing on education on the web. I invte the authors to consider the addition of examples to accompany each formula and to better justify all their design choices.
- The experimental results highlight that the proposed approach grings gains with respect to several baselines. However, the reported gains, often at the second or third decimal, leave room for questions regarding the real impact on learners. Moreover, the authors merely focus on machine-learning oriented, offline metrics, without any connection to how this can enable a better learning on the Web. To show the significance of the improvement, it is suggested that the authors consider an online evaluation with A/B testing to provide a more practical context.

Overall, embracing the suggestions to diversify datasets, enhance methodological accessibility, and contextualize numerical gains within practical learning scenarios can elevate the paper's impact and resonance within the scientific community.

**Questions:**

- Why did you share only a sample of the data?
- Why only one dataset has been considered?

**Reviewer Confidence:**

4: The reviewer is certain that the evaluation is correct and very familiar with the relevant literature

**Scope:**

2: The connection to the Web is incidental, e.g., use of Web data or API

---

### Official Review · Reviewer_wc71 · 2023-12-01

**Novelty:** 5
**Technical Quality:** 4

**Review:**

This paper analyzes the challenges of online knowledge management platforms and proposes CAHL for predicting the future knowledge acquisition of employees. The proposed method is well-crafted and reasonable. However, certain aspects of the Employee-Knowledge Graph Construction and experiments lack clear presentation or explanation.

Advantages:
1. The paper is well-structured and the presentation, and chart are easy to follow.
2. The proposed CAHL is well-crafted and reasonable, and I especially appreciate the part about capturing the co-occurrence relationships and prerequisite relationships between job knowledge.
3. Related work about different aspects involved in the proposal.

Weakness:
1. The experimental part of the article is not sufficient. It only conducts experimental analysis on one data set, which is not convincing enough.
2. The Problem Definition of knowledge development is not sufficient. (especially for Employee-Knowledge Graph Construction)

**Questions:**

1. Why can the Employee-Knowledge Graph be constructed in this way? How to get prerequisite relationship and association prompt relationship between knowledge.
2. Is there any other analysis and experiments on other datasets to verify the effectiveness of the method?

**Reviewer Confidence:**

3: The reviewer is confident but not certain that the evaluation is correct

**Scope:**

4: The work is relevant to the Web and to the track, and is of broad interest to the community

---

### Decision · Program_Chairs · 2024-01-22

**Decision:**

Accept (Oral)

**Comment:**

This paper proposed an approach for modeling hybrid learning for predicting job knowledge development. The collaboration-aware hybrid learning approach incorporates a job knowledge embedding module to capture inherent rules, an employee embedding module for employee representations, and a hybrid learning simulation module for predicting future-acquired job knowledge. Empirical results on a real-world dataset show the proposed approach works well in addressing these complexities.